# Morphometric convergence between Proterozoic and post-vegetation rivers

Alessandro Ielpi[1], Robert H. Rainbird[2], Dario Ventra[3,4] & Massimiliano Ghinassi[5]

Proterozoic rivers flowed through barren landscapes, and lacked interactions with macroscopic organisms. It is widely held that, in the absence of vegetation, fluvial systems featured barely entrenched channels that promptly widened over floodplains during floods. This hypothesis has never been tested because of an enduring lack of Precambrian fluvial-channel morphometric data. Here we show, through remote sensing and outcrop sedimentology, that deep rivers were developed in the Proterozoic, and that morphometric parameters for large fluvial channels might have remained within a narrow range over almost 2 billion years. Our data set comprises fluvial-channel forms deposited a few tens to thousands of kilometres from their headwaters, likely the record of basin- to craton-scale systems. Large Proterozoic channel forms present width:thickness ranges matching those of Phanerozoic counterparts, suggesting closer parallels between their fluvial dynamics. This outcome may better inform analyses of extraterrestrial planetary surfaces and related comparisons with pre-vegetation Earth landscapes.

[1] Harquail School of Earth Sciences, Laurentian University, Sudbury, Ontario, Canada P3E 2C6. [2] Geological Survey of Canada, Ottawa, Ontario, Canada K1A 0E8. [3] Department of Earth Sciences, University of Geneva, CH-1205 Geneva, Switzerland. [4] Faculty of Geosciences, Utrecht University, 3583CS Utrecht, The Netherlands. [5] Department of Geosciences, University of Padua, 35131 Padua, Italy. Correspondence and requests for materials should be addressed to A.I. (email: aielpi@laurentian.ca).

The interpretation of ancient sedimentary environments relies on comparisons between their rock records and the dynamics of modern sedimentary landscapes. Pre-Ordovician terrestrial realms were devoid of macroscopic life forms[1,2]. Accordingly, the study of their fluvial sedimentology has long been plagued by the scarcity of modern analogues, where continental surfaces are greatly affected by biota, principally vegetation. A lack of published data on pre-vegetation channel geometry has contributed to the formulation of a non-actualistic hypothesis that portrays wide, shallow pre-vegetation rivers incapable of incising deep channels with narrow hydraulic profiles[3,4]. This has been ascribed to the lack of sediment binding by plant roots and to the cohesionless nature of barren substrates[5]. Status quo is that this inferred fluvial style bears some degree of resemblance to ephemeral and low-sinuosity streams of modern dryland environments[6,7], and a widely held notion maintains that most pre-vegetation rivers were subject to ephemeral or seasonal discharge regimes, independently of climate[3]. This hypothesis contrasts with recent numerical modelling and remote-sensing data from large outcrops, which reveal a greater tendency towards deeper channel forms than previously portrayed[8,9]. The occurrence of deep channels on unvegetated planets such as Mars and Saturn's moon Titan also indirectly indicates that barren landscapes may sustain stable fluvial conduits[10,11]. This uncertainty in reconstructing fundamental, ubiquitous elements and processes of subaerial landscapes hinders our ability to understand Earth's geomorphic evolution through the longest timespan of its history, as well as the evolution of other planetary surfaces. This is even more striking considering that many Proterozoic basins host multi-kilometre-thick sedimentary fills, with comparable stratigraphic completeness to Phanerozoic ones[3].

Here we demonstrate that during a significant portion of the Proterozoic, large-scale fluvial systems had channels that shared morphometric characteristics with those of post-vegetation and modern rivers, irrespective of their planform or position within sedimentary basins. Our results disprove the widespread inference that all ancient unvegetated rivers were wide and shallow. This hypothesis is supported by a data set of width and thickness values from 156 fully preserved channel forms ranging in age from 1.9 to 1.0 Ga, once part of both proximal and distal drainage networks (Fig. 1). Analysis of remotely sensed channel geometries within extensively exposed basin tracts, coupled with successive ground surveys, represents a powerful tool for reconstructing the morphology and assessing the hydrodynamics of ancient rivers[12,13]. This approach is here used to broaden the range of known pre-vegetation alluvial forms.

## Results

**Recognition of alluvial channel forms**. The recognition of channel forms is critical in fluvial sedimentology, and relies on the detection of prominent erosional scours overlain by sediment fills related to either stages of active discharge or channel abandonment. Active-channel fills typically contain gravelly and sandy deposits pointing to bedload traction by waterflows, while abandoned-channel fills consist of finer-grained, sandy and muddy deposits settled in slough waters after flow diversion to another conduit[14]. Active channels are either mobile or fixed. Mobile channels adjust their hydraulic section and position over floodplains through successive episodes of incision, bed aggradation, bank migration and avulsion. Mobile channels capable of significant lateral migration and concomitant vertical aggradation will generate amalgamated sediment bodies larger than their original hydraulic section at any given time[15]. In such cases, the geometry of individual abandonment fills preserved within the resultant sediment body approximates the instantaneous hydraulic section at the time of channel diversion. By contrast, fixed channels adjust their hydraulic section without significant lateral shift, and their geometry is considered a valid approximation of the alluvial cross-section during the last effective flood[16]. Outcrop studies can typically resolve channel forms no wider than a few hundred metres, a limitation that introduces a significant bias towards small-scale fluvial systems. On the other hand, remotely sensed images proved critical for the identification of Phanerozoic fluvial forms at larger scales

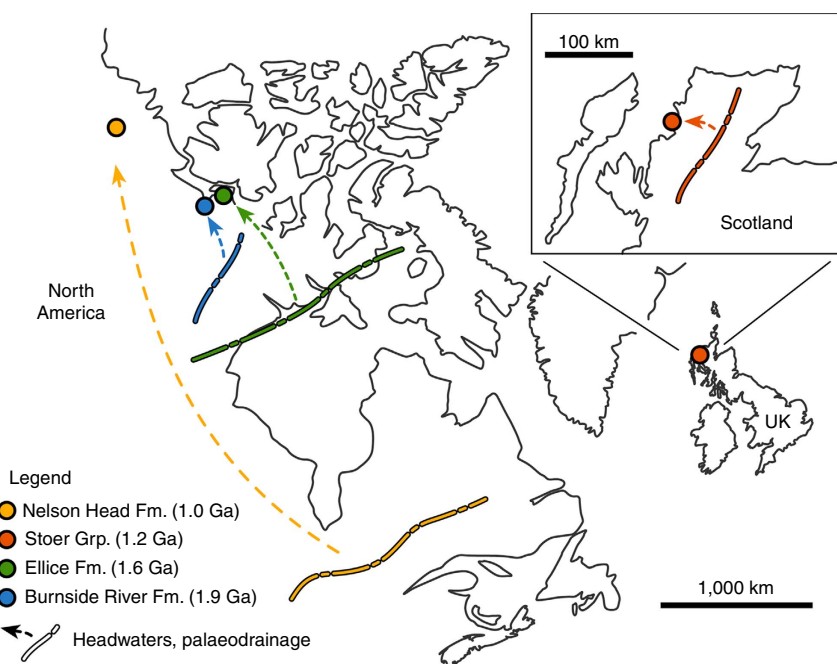

**Figure 1 | Location and palaeogeography of the data set.** Relative position of the considered rock units in an approximate reconstruction of the Neoproterozoic North Atlantic region[50], with approximate position of headwaters and palaeodrainage vectors reported.

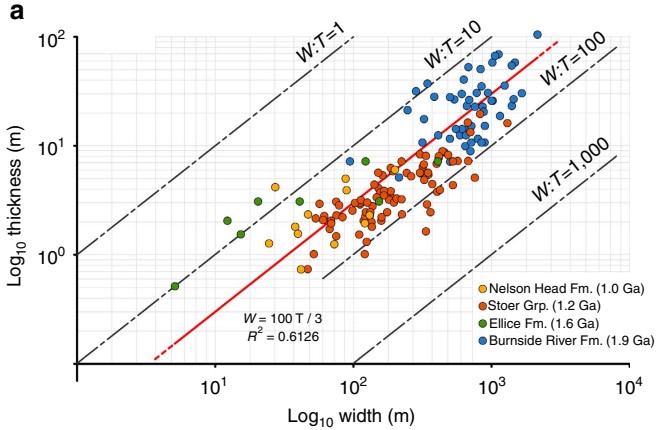

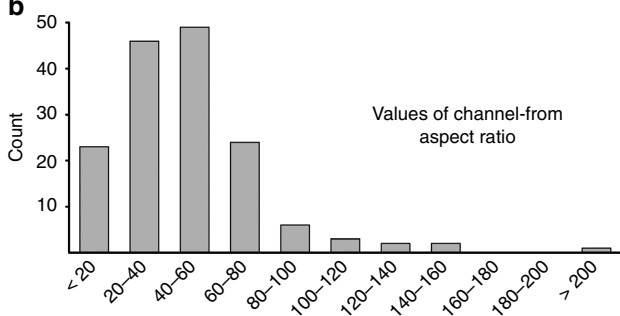

**Figure 2 | Morphometry of exposed Proterozoic channel forms.**
(**a**) Logarithmic width–thickness plot reporting the database of 156 Proterozoic channel forms (Supplementary Data 1). Linear regression line is reported in red. (**b**) Histogram showing relative frequencies of values of aspect ratio in the database shown in **a**.

(that is, several kilometres), more likely to represent extensive, basin-scale systems[12,13].

**Proterozoic alluvial-channel morphometry.** Our data set includes active fixed-channel forms, active mobile-channel forms and channel-abandonment intervals that belonged to fluvial systems capable of developing alluvial cross-sections several hundred metres to a few kilometres wide (Figs 2 and 3), scales that currently typify some of the largest rivers on Earth (for reference, the largest modern rivers by discharge have trunk channels typically 1–5 km wide[17]). The disambiguation between channel forms and the fill of incised valleys is critical in this context, and was based on the following criteria[15,18]: channel forms are floored by scours that are neither part of, nor incised onto, wider surfaces traceable across interfluves; channel forms have thickness comparable to the vertical relief of erosional reactivations contained therein; and channel forms are 10–100 times narrower than recognized incised valleys.

Our data set (Supplementary Data 1) is compiled from: the 1.9 Ga Burnside River Formation of Kilohigok Basin (Nunavut, Canada), which comprises active-channel fills in a foreland basin at an estimated distance of ∼120 km from their headwaters[19]; the 1.6 Ga Ellice Formation of Elu Basin (Nunavut), exposing active-channel fills in an intracratonic basin ∼1,000 km from their source area[20]; the 1.2 Ga Stoer Group, part of the classical Torridonian succession of Scotland (UK), containing active-channel fills in a rift basin no farther than 60 km from their headwaters[21]; and the 1.0 Ga Nelson Head Formation of Amundsen Basin (Northwest Territories, Canada), exposing

abandoned-channel fills in an intracratonic basin at the distal end of a pan-continental drainage system that extended over 3,000 km (ref. 22). Sedimentologic details[9,23–25] indicate that these stratigraphic units consist mostly of cross-stratified sandstone with a narrow range in thickness for cross-strata sets (generally 20–70 cm), implying a correspondingly restricted discharge range within subcritical flow regimes during channel-bed aggradation[26,27]. Coupled with the scarcity of plane beds and supercritical-flow structures, this points to fluvial systems not frequently subject to sharp discharge variations, but rather characterized by relatively constant flow typical of perennial regimes[28]. Channel fills are part of sediment assemblages also including fluvial sand-sheets, overbank fines and strata accumulated in aeolian ergs. Fluvial sand-sheets are composed of stacked bedsets exhibiting compound cross-bedding, interpreted as bar amalgamation after migration of mobile channels several metres deep[9,23,24].

The aspect ratios (width:thickness) of observed channel forms are shown in logarithmic plots in Fig. 2a, and the salient features of each rock unit (for example, channel clustering, proximity to basement relief, geometry and inferred sinuosity) are reported in Table 1. The geometry and architecture of channel deposits at the remote and outcrop scale are summarized in Figs 3 and 4. Overall, the aspect ratio data set displays a positively skewed, Gaussian distribution with most values between 20 and 60 (Fig. 2b). A linear regression derived from the entire data set indicates that:

$$W = 100\,T/3 \pm 29\,T \qquad (1)$$

where $W$ and $T$ represent width and thickness of channel forms, respectively, and $\pm 29$ indicates the data set's s.d. ($R^2 = 0.6162$). Since, by definition, a channel form must have values of $W$ and $T$ greater than zero, the linear regression includes the $W = 0$, $T = 0$ point (Fig. 2a).

Roughly 85% of the data set is represented by channel forms hundreds to thousands of metres wide from the Burnside River Formation and Stoer Group (Fig. 2a and Table 1). There, channel forms are grouped in clusters located within prominent basement palaeo-valleys (Fig. 3a,c), and include low-sinuosity, active-channel fills (Fig. 4a,c). In both the Burnside River Formation and Stoer Group, extensive erosional surfaces, wider than any individual channel form, indicate that rivers flowed through large valleys, both alluvial and incised on bedrock[9,25]. These features point to entrenchment and clustering[29] of large alluvial channels incapable of significant lateral mobility within drainage systems whose position was significantly confined by basement topography[30]. These channel clusters were preserved from a few tens to about a hundred kilometres from their headwaters[19,21], suggesting relatively proximal alluvial tracts. Channel forms are significantly thicker than the tallest bar slip-faces preserved therein (Table 1), indicating a dominant bed aggradation during the active-channel lifespan[31], or that the observed channel-fill thickness exceeded the original channel depth at any one time. Since lateral migration was limited, so was channel mobility, and the width of preserved channel forms is thus considered a valid approximation of original channel width[16].

Channel forms tens to hundreds of metres wide (Fig. 2a and Table 1) in the Ellice and Nelson Head formations do not occur in clusters and include both active- and abandoned-channel fills (Fig. 4b,d). Inferred planform styles point to low/intermediate sinuosity, typical of rivers capable of bank migration. These units were preserved ∼1,000–3,000 km from their headwaters[20,22], suggesting broad, distal alluvial plains on which channels shifted or avulsed over significant distances[32]. Absence of channel clusters and rare channel-fill interconnection (Fig. 3b,d)

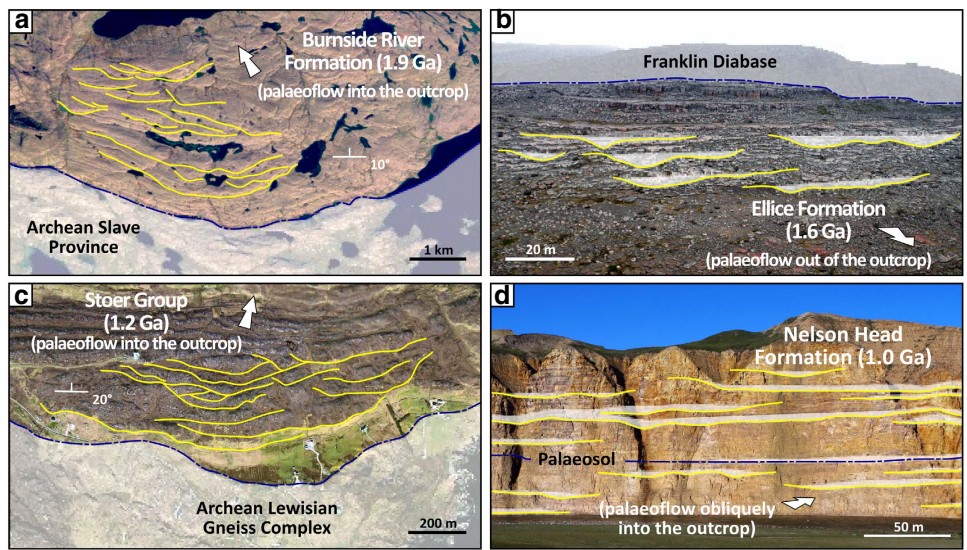

**Figure 3 | Remote sensing of Proterozoic channel forms.** (**a**) Satellite view of the 1.9 Ga Burnside River Formation: narrow channel cluster found in proximity to basement topography. (**b**) Oblique aerial photopanel of the 1.6 Ga Ellice Formation: isolated channel forms contained in amalgamated fluvial–bar complexes. (**c**) Satellite view of the 1.2 Ga Stoer Group: broad channel cluster within a palaeovalley carved on basement bedrock. (**d**) Cliff exposure of the 1.0 Ga Nelson Head Formation: abandoned-channel fills engulfed in amalgamated fluvial–bar complexes. Basal channel surfaces reported with yellow lines.

**Table 1 | Salient features of the rock units considered in this study.**

| Rock unit, age | km from headwaters | No. of data, type of fill | Degree and type of clustering | Observed basement relief in the outcrop belt | Range of width (m) | Range of thickness (m) | Range of aspect ratio | Maximum bar slip-face height | Inferred sinuosity index |
|---|---|---|---|---|---|---|---|---|---|
| Burnside River Formation, 1.9 Ga | ~120 | 52 Active-channel fills | Channel fills forming narrow clusters | Up to 250 m over 15 km along strike | 92–1,106 | 5–102 | 9–83 (mean = 37) | 25 m | Low (<1.25) |
| Ellice Formation, 1.6 Ga | ~1,000 | 8 Active-channel fills | No apparent clustering | Up to 30 m over 20 km along strike | 5–400 | 0.5–7 | 6–57 (mean = 21) | 4 m | Low to intermediate (1.25–1.5) |
| Stoer Group, 1.2 Ga | ~60 | 84 Active-channel fills | Channel fills forming narrow to broad clusters | Up to 400 m over 3 km along strike | 46–1,276 | 0.7–19 | 20–205 (mean = 59) | 15 m | Low (<1.25) |
| Nelson Head Formation, 1.0 Ga | ~3,000 | 12 Abandoned-channel fills | No apparent clustering | Absent | 24–196 | 0.7–60 | 7–63 (mean = 34) | N/A | Low to intermediate (1.25–1.5) |

indicate either anabranching river planforms or the distal sectors of distributive fluvial systems, both commonly represented by relatively narrow channel fills dispersed within dominant finer-grained alluvial deposits[33,34]. Channel entrenchment and sinuosity may have been enhanced by the local occurrence of cohesive palaeosol horizons (Fig. 3d), which provided bank stability[35], or by confinement betwgeen aeolian landforms[36]. Channel forms in the Ellice Formation have thicknesses comparable to that of the tallest bar slip-faces observed therein (Table 1), indicating limited bed aggradation, as expected in mobile channels capable of lateral migration[15]. This implies that their width may be significantly greater than the original alluvial cross-section at any given time. Finally,

the morphometry of channel-abandonment fills in the Nelson Head Formation provides a solid approximation for original channel geometry.

**Comparison with vegetated fluvial systems.** Direct comparison of channel morphometry represents a powerful tool to infer depositional dynamics of ancient fluvial systems[5,15,37]. A comparative framework of 156 reported Proterozoic channel forms and vegetated counterparts, both ancient and modern (Fig. 5), is discussed here. The Proterozoic channel forms are representative of individual channels that, despite variable degrees of vertical aggradation and lateral adjustment, show no evidence

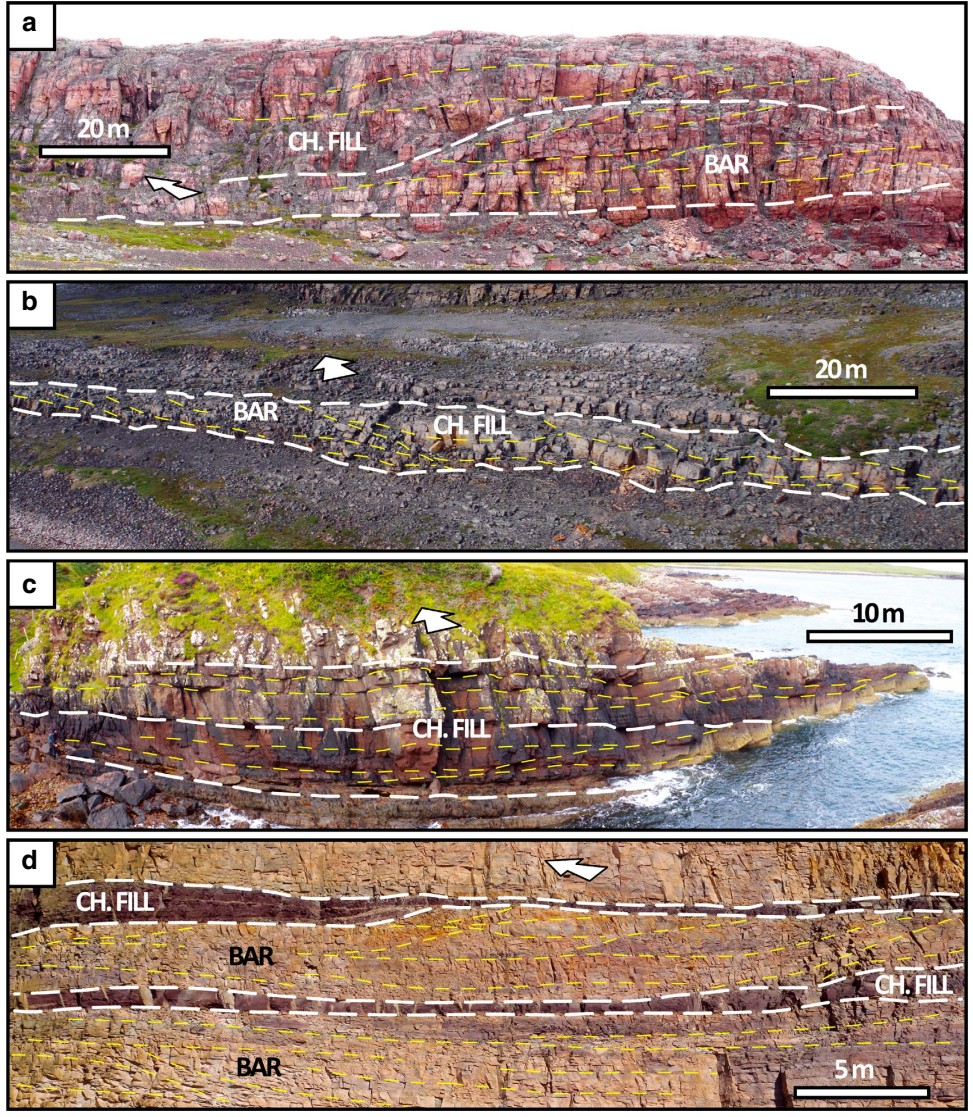

**Figure 4 | Field aspects of Proterozoic channel forms. (a,b)** Active-channel fills sided by bank-attached bars in the Burnside River (**a**) and Ellice (**b**) formations. The geometry of the channel fills (when entirely observable from satellite) is considered as an approximation of the alluvial section during the last major flood. (**c**) Stoer group: aggradational, active-channel fill containing low-relief in-channel bars. Despite prominent bed aggradation, the tallest bar slip-face (∼4 m) provides a first-order approximation of maximum channel depth. (**d**) Nelson Head formation: red-coloured abandoned-channel fills attached to mid-channel and bank-attached bars (tan-coloured). The upper red unit is part of a fully preserved channel form; the lower red unit is scoured by the overlying deposits and is not considered in the data set.

of amalgamation during aggradation. As such, comparative data from ancient vegetated landscapes (Late Ordovician to Pleistocene) are similarly selected from rock units that fully preserve non-amalgamated channel forms (Supplementary Data 2 and Supplementary Figs 1–4). Published compilations, although more exhaustive[15,38,39], cannot be used since they refer to mixed data from both individual and amalgamated channel forms. Data on ancient vegetated systems are derived from abandoned and active channel fills, and belong to 29 stratigraphic units grouped into four subpopulations: mobile braided channels; mobile channels with intermediate sinuosity; fixed and mobile channels with meandering anastomosing planforms; and fixed and mobile channels belonging to distributary systems, that is, deltas, fluvial piedmonts and fans. Collective envelopes[15] constructed in logarithmic, width-thickness plots, show that Ordovician to Pleistocene channel forms range from $10^0$ to almost $10^3$ in aspect ratio (Fig. 5b). Comparison between

Proterozoic and Phanerozoic channel width-thickness spaces shows significant overlap (Fig. 5a,b), that is, Proterozoic channel forms are neither wider nor thinner than younger counterparts from vegetated landscapes. This is valid irrespective of the fact that Proterozoic values are largely representative of low-sinuosity channels, while Phanerozoic values represent a range of planview styles. Proterozoic end-member values from the Burnside River Formation even show thicker, and thus comparatively narrower, profiles than post-vegetation channel forms (Fig. 5a). This marked overlap in channel geometry represents circumstantial evidence that large rivers established on unvegetated landscapes generated channel cross-sections with low width:depth ratio.

Comparisons with Holocene and modern alluvial morphometries are drawn from 192 cross-sections of alluvial rivers drawn from current literature (Supplementary Data 3 and Supplementary Fig. 5), grouped into two subpopulations of low-sinuosity

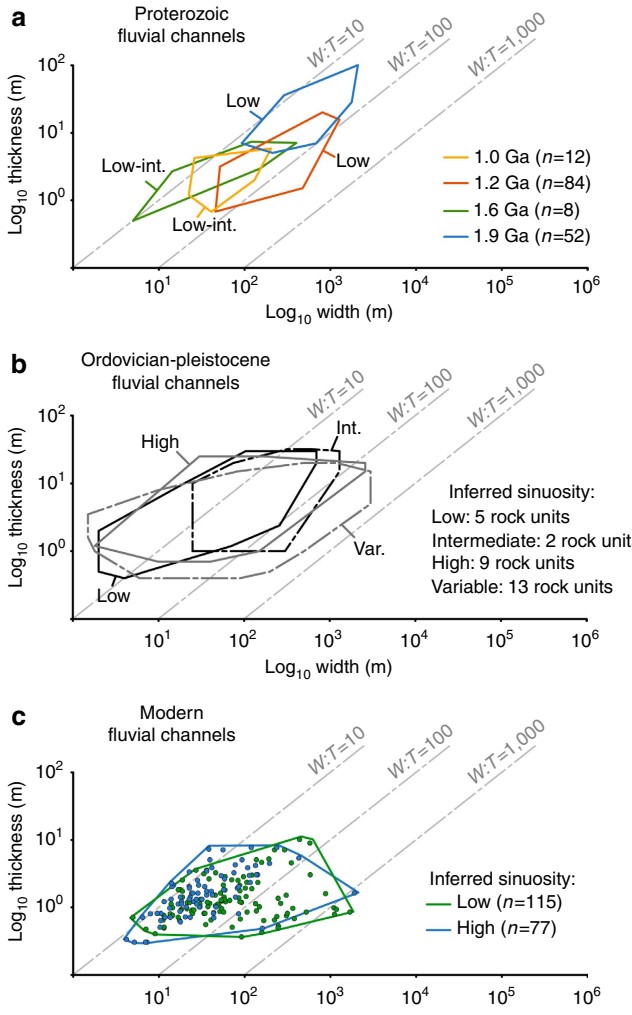

**Figure 5 | Morphometric comparisons between pre- and post-vegetation channel forms.** (**a**) Collective envelopes[12] constructed from the database of 156 channel forms. Inferred sinuosity of each unit is reported. (**b**) Collective envelopes for Ordovician to Pleistocene channel forms found in 29 rock units worldwide (Supplementary Data 2 and Supplementary Figs 1–4), grouped based on inferred sinuosity. (**c**) Data points and collective envelopes for 192 modern alluvial cross-section measures along perennial rivers not modified by anthropic activity (Supplementary Data 2 and Supplementary Fig. 5), grouped based on sinuosity.

and high-sinuosity channels. Following the approach taken by previous compilations[40], only data from rivers not substantially modified by anthropic activity were compiled. When comparing modern alluvial cross-sections to those of ancient channel bodies, the effect of sediment compaction in the latter must also be accounted for. While accurate figures of alluvium compaction cannot be retrieved, it has been suggested that sand-rich Precambrian alluvial deposits yield compaction rates of ∼10% (ref. 41). Notwithstanding this uncertainty, Proterozoic channel forms have widths fully overlapping with the morphometric data set of modern channels, which show an aspect ratio range of $10^1$–$10^3$ irrespective of sinuosity (Fig. 5a,c). This result again supports the inference that pre-vegetation alluvial cross-sections were not significantly wider than vegetated ones. Strong overlap also exists between the depth of modern channels and the thickness of distal Proterozoic channel forms (Fig. 5a,c), confirming that bed aggradation in the latter was somewhat

limited. In addition, proximal Proterozoic channel forms exhibit thickness values much higher than the depth of modern channels (Fig. 5a,c). This again suggests enhanced channel-bed aggradation for proximal Proterozoic alluvial tracts and the generation of channel fills thicker than original channel depths.

In synthesis, remotely sensed images and field data from a broad range of fully preserved Proterozoic fluvial-channel fills and comparisons with Phanerozoic channel morphometric data clearly show that commonly held assumptions on invariably shallow and wide pre-vegetation fluvial channels are untenable. Since at least 1.9 Ga, perennial alluvial channels belonging to large-scale fluvial systems were able to develop deep hydraulic profiles, suggesting channel–overbank relationships matching those observed in post-Ordovician landscapes.

**A revised model for Proterozoic rivers.** Our evaluation of fluvial channel form and hydrodynamics in Proterozoic terrestrial landscapes stems from a novel approach based on remote-sensing observations of channel fills from extensive outcrop belts, at such scales that their morphometric attributes could not be fully appreciated by means of conventional ground surveys. Previous attempts to reconstruct Precambrian fluvial-channel geometries were based on outcrops a few hundred metres wide[9], a scale that only allows for the resolution of comparatively small fluvial forms. Hence, it appears that extant models for Precambrian rivers are biased towards rather small systems. The significance of Proterozoic alluvial bodies related to sheet-like alluvial cross-sections is de-emphasized in this study, and we stress that analysis of remotely sensed kilometre-scale exposures is critical for characterizing the geometry of larger fluvial systems. Our results do not rule out *a priori* the existence of broad, shallow rivers in the Proterozoic, rather they expand the range of known fluvial forms, considering those at larger scales likely representative of basin- to craton-scale drainages. The investigated channel forms consistently reveal relatively narrow cross-sections through palaeogeographic settings ranging from proximal to distal alluvial tracts. The occurrence of large bar forms (4–25 m thick) and the lack of sheeted channel fills dismiss the hypothesis that channel deposits aggraded within broad, shallow, highly mobile stream-beds. The linear correlation derived from our data set ($W = 100\,T/3 \pm 29\,T$) provides a first-order approximation of the width of deep Proterozoic channel forms that are only partially preserved or exposed.

Our data set comprises the products of large fluvial systems originating in extensive and long-lived orogenic belts (Fig. 1). It is argued that the deep and stable channel forms discussed here might have resulted from protracted tectonic stability during times of supercontinent amalgamation[42]. Channelization in proximal alluvial tracts was favoured by drainage funnelling along valley thalwegs, where rivers assembled through time forming narrow clusters. These fluvial systems were likely subject to prominent bed aggradation, as expected along fluvial tracts confined by steep basement topography. The role of bank cohesion in these proximal tracts may have been minimal or absent. Conversely, channelization in more distal, low-gradient alluvial settings might have been controlled by a combination of stable discharge and bank cohesion. A tendency towards increasingly stable discharge can be postulated for larger systems; this because such systems likely collected runoff from wide catchments that fed significant year-round base flow, and were thus less sensitive to seasonal climates and extreme precipitation events. Previous studies on Precambrian fluvial landscapes pointed to the relevance of bank strengthening by mud cohesion, soil crusts, microbial mats[2,43], ground ice[44], early alluvium cementation[45] or spatial confinement induced by aeolian landforms[36]. Clay-rich alluvial substrates were relatively

uncommon on pre-vegetation landscapes compared to the present, lacking the pervasive penetration and enhanced (bio)chemical weathering by root systems[6]. Nonetheless, bank strengthening over distal alluvial plains was probably enhanced by reduced gradients, to the point where incipient substrate cohesion would even have favoured the development of sinuosity[23,24,46].

It is still widely accepted that the early Palaeozoic rise of vascular land plants contributed to a significant diversification of fluvial planform styles[5]. However, comparisons of Proterozoic, Devonian to Pleistocene and modern channel morphometries suggest that the formative alluvial cross-sections of deep channels belonging to large fluvial systems did not vary significantly over a range of planform styles for almost 2 billion years, including an interval of ~1.6 billion years before the early evolution and diffusion of vascular plants. These results are consistent with our understanding of the Proterozoic Earth by 2.0 Ga, where cratonic sedimentary basins displayed source-to-sink patterns of sedimentation and accumulation not too dissimilar from modern ones[3]. Data reported here counter the widespread notion that pre-vegetation alluvial rivers were *uniquely* characterized by wide, shallow, frequently shifting channels of high aspect ratio, and thus we contend that parallels commonly drawn between Proterozoic fluvial systems and modern ephemeral rivers are not warranted. The perennial regime inferred from the observed channel deposits is related to stable patterns of discharge relative to sediment supply. Sediment production and delivery might have been relatively steady in the Proterozoic, with bedrock relief exposed to vigorous physical and acidic chemical weathering[47,48], and steady sediment yield in the absence of binding vegetation along upland slopes. This granted more continuous sediment delivery from large, well-integrated catchments that fed fluvial systems of conspicuous extent, such as those recognized from stratigraphic records of the Canadian Shield[9,19,22–24,32].

**Outlook.** A number of Precambrian fluvial units are exposed over vast and still underexplored intracratonic basins worldwide, and we expect that further advances gained from their study will expand the data set discussed here. Future remote-sensing analyses may adjust the morphometric parameters established here and will provide additional information on Archean to early Palaeoproterozoic fluvial-channel forms to further test hypotheses of evolving or invariant channel morphometry through deep time. Our data provide evidence that current hypotheses and conceptual models on the evolution of fluvial systems through deep geological time need revision. Hence, we call for a critical reassessment of the environmental controls on fluvial-channel patterns and morphometry. Once systematic, highly resolved morphometric compilations on alluvial channels from other planetary bodies[49] become available, our data set may also inform future comparisons between terrestrial and extraterrestrial fluvial systems.

## Methods

**Outline.** This study is based on a quantitative data set of width and thickness for Proterozoic channel forms (Fig. 2) identified from satellite imagery and photo-panels acquired during helicopter flybys. Only channel forms with fully preserved margins were considered, that is, those clearly separated from adjoining deposits and not truncated by overlying erosional surfaces (Fig. 3). Channel forms were ground-checked during GPS-aided field traverses. Values of apparent channel width were trigonometrically corrected to cross-sections oriented normal to local palaeoflow and apparent thickness values were corrected for tectonic dip (Supplementary Data 1).

**Data availability.** The authors declare that all data supporting the finding of this research are available within the paper and its Supplementary Information.

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

## Acknowledgements

This research was supported by a Discovery Grant to A.I. from the Natural Sciences and Engineering Research Council of Canada (NSERC), and in-kind support from the Canadian Northern Economic Development Agency's (CanNor) Strategic Investments in Northern Economic Development (SINED), and Natural Resources Canada's (NRCan) Geomapping for Energy and Minerals Program, phase 2 (GEM-2). We thank two anonymous reviewers for their insightful comments.

## Author contributions

A.I. and R.H.R. jointly conceived and undertook the studies and fieldwork in Arctic Canada. A.I., M.G. and D.V. jointly conceived and undertook the study and fieldwork in Scotland. All authors contributed to the writing of the manuscript. Literature compilations were undertaken by A.I.

## Additional information

**Competing interests:** The authors declare no competing financial interests.

