## [Peer Review File · Nature Communications]

Reviewers' Comments:

Reviewer #1 (Remarks to the Author)

The paper uses a number of case studies to demonstrate, fairly convincingly, that Proterozoic rivers include examples that had deeply incised channels. The authors claim that this goes against received wisdom and that pre-vegetation rivers should, according to a broad consensus, be characterized by broad shallow channels. This consensus does indeed exist, although it also holds that such pre-vegetation rivers were braided (sheet-braided systems) and never (hardly ever?) meandering, which is a key point neglected by the authors. I also feel (and this is generally neglected in this type of literature) that the deep-time fluvial archive is generally an imperfect record of the types of riverine environment that existed. This is because only lowland fluvial systems that occupy subsiding depocentres have any chance at all of their sediments surviving to be part of the long-timescale geological record. This would tend to favour preservation of lowland broad aggradational systems.

Nonetheless, the impression given in the manuscript that this claim of deeply incised Proterozoic river channels is entirely novel is false, as is hinted by numbered references 8 and 9, which provide more in-depth discussion of the topic, including one of the case studies in the present paper (the Burnside River Formation of Arctic Canada), with one of the present authors a contributor.

That does not mean that this is not a valuable and concise review of the topic, although it might raise a question about whether this journal is where it belongs.

Reviewer #2 (Remarks to the Author)

This is an interesting study that contains plenty of supporting evidence for the case that Proterozoic rivers were not that different from Phanerozoic ones. This would appear to have important implications for the criteria which fluvial sedimentologists use in their discussions of rivers of various types and various ages.

I have maintained in my recent book "Fluvial depositional systems" that the data compiled by Gibling (2006) show that the range of width-depth ratios of braided versus meandering rivers define similar envelopes, suggesting that the broad architectural dimensions of these two archetypes are not that different. This paper extends the analysis to rivers of all types of Proterozoic age.

What we are seeing here is the challenging of stereotypes that go back to the kinds of comments made by Schumm (1968), Cotter (1978), Long (1978), etc. The stereotypes come with arguments that seemed quite logical at the time, but this paper suggests that they are wrong. Here are a couple of possible reasons why Ielpi's data might be different:

1) The outcrop data base for Proterozoic rivers has been biased to rather small rivers, which means that there may have been few opportunities to see actual channel margins, leading to the idea that perhaps channel margins were not part of the picture, i.e., the rivers were sheet-like in form. The presentation in this paper of aerial and satellite imagery of unquestionably large rivers certainly shows channel forms, and this is the basis of their argument for a new model.

2) Is it possible that all we are seeing here is the record of large rivers that are entrenched, with distinct concave-up channel forms, because they represent the long-lived stability that comes with regional tectonic control? Rivers like the Mississippi, Amazon, etc., have occupied much the same position for millions of years because of regional tectonic control. Smaller rivers, of whatever age, and in any climatic setting, might be freer to meander or avulse, and in such smaller rivers, perhaps the architectural differences caused by bank stability, etc., that Gibling, Davies, Long and

others have attributed to the development of vegetation and changes in sediment load, are more likely to leave their mark on channel form.

These are just speculations, which the authors might want to consider. Apart from this I have no substantive comments to make on a very interesting and well documented study.

Reviewer 1 – Anonymous

R1.1 > The paper uses a number of case studies to demonstrate, fairly convincingly, that Proterozoic rivers include examples that had deeply incised channels. The authors claim that this goes against received wisdom and that pre-vegetation rivers should, according to a broad consensus, be characterized by broad shallow channels. This consensus does indeed exist, although it also holds that such pre-vegetation rivers were braided (sheet-braided systems) and never (hardly ever?) meandering, which is a key point neglected by the authors.

Answer > A stimulating comment, notwithstanding the fact that our research actually has a more direct focus on channel morphometry rather than planform style. To address this comment, we added key sentences at **lines 46, 175-176**, and **245-249** to remark that our results apply to fluvial systems ranging from low to intermediate sinuosity. At **lines 175-176** we also acknowledge that Proterozoic fluvial systems had generally lower sinuosity than their post-Ordovician counterparts.

R1.2 > I also feel (and this is generally neglected in this type of literature) that the deep-time fluvial archive is generally an imperfect record of the types of riverine environment that existed. This is because only lowland fluvial systems that occupy subsiding depocentres have any chance at all of their sediments surviving to be part of the long-timescale geological record. This would tend to favour preservation of lowland broad aggradational systems.

Answer > We agree with the reviewer that, in principle, the sedimentary record may be imperfect given specific circumstances, and that subsidence is paramount for its long-term preservation. However, this is valid irrespective of the geological timespan considered (i.e., not solely for the Proterozoic), and most of all, irrespective of what depositional systems are considered (i.e., not solely fluvial).

More in the specific, the notion that the Proterozoic continental rock record is patchier than the Phanerozoic one is based on the common assumption that – in a linear fashion – the deeper the geological time, the less complete its sedimentary record is. The comprehensive review by Eriksson and co-authors (cited in our manuscript as 3: Eriksson et al., 1998) was among the first to refute this concept, pointing to the fact that many Precambrian continental basins were likely characterized by favourable budgets of accommodation relative to sediment supply, allowing for the remarkable completeness of their sedimentary record.

To address this point, we remark, early in the manuscript (at **lines 41-43**), that many preserved Precambrian basins host multi-kilometre-thick packages of fluvial strata, arguably with a similar (if not greater) stratigraphic completeness than Phanerozoic ones (reference to the work of Eriksson et al., 1998, is also reported here). Later in the text, at **lines 263-265**, we state again that many thick Precambrian fluvial successions are preserved and exceptionally exposed worldwide.

R1.3 > Nonetheless, the impression given in the manuscript that this claim of deeply incised Proterozoic river channels is entirely novel is false, as is hinted by numbered references 8 and 9, which provide more in-depth discussion of the topic, including one of the case studies in the present paper (the Burnside River Formation of Arctic Canada), with one of the present authors a contributor. That does not mean that this is not a valuable and concise review of the topic, although it might raise a question about whether this journal is where it belongs.

Answer > That the current research stems from a unique collection of case studies is made explicit in several instances throughout the paper (for example, at **lines 92-100**). Such previous case studies only described single stratigraphic units and, in doing so, they could not in principle provide wider implications of their results. To provide the latter is the main intention of this manuscript.

While a few recent studies discuss individual occurrences of deeply channelled fluvial systems in the Proterozoic, none has summarized and discussed their morphometric features as a whole in a separate research. The reference to the work of Almeida et al. (2016) remains appropriate in this context, although their work is only partly based on outcrop evidence, while rather it heavily relies on numerical modelling. The novelty of the current research is again that, for the first time, an integrated database of Proterozoic channel morphometry from several rock units worldwide is presented, discussed, and corroborated by comparisons with published literature on post-vegetation channels.

Reviewer 2 – Anonymous

R2.1 > The outcrop data base for Proterozoic rivers has been biased to rather small rivers, which means that there may have been few opportunities to see actual channel margins, leading to the idea that perhaps channel margins were not part of the picture, i.e., the rivers were sheet-like in form. The presentation in this paper of aerial and satellite imagery of unquestionably large rivers certainly shows channel forms, and this is the basis of their argument for a new model.

Answer > A very good point that was only briefly mentioned in the original submission, and that has now been emphasized. At **lines 224-227**, we remark that previous reconstructions were only based on outcrops not representative of entire fluvial forms; hence, they have only provided a point of view biased towards smaller systems. Contrarily, remotely sensed imagery used in the present work reveals the geometry of fluvial forms likely representative of continental-scale drainages (e.g., **lines 82-86, 212-215**).

R2.2 > Is it possible that all we are seeing here is the record of large rivers that are entrenched, with distinct concave-up channel forms, because they represent the long-lived stability that comes with regional tectonic control? Rivers like the Mississippi, Amazon. Etc., have occupied much the same position for millions of years because of regional tectonic control. Smaller rivers, of whatever age, and in any climatic setting, might be freer to meander or avulse, and in such smaller rivers, perhaps the architectural differences caused by banks stability, etc., that Gibling, Davies, Long and others have attributed to the development of vegetation and changes in sediment load, are more likely to leave their mark on channel form.

Answer > Another good point. We agree that long-term stability in the position of alluvial trunks – something often ascribed to tectonic control – may have also contributed to the establishment of large, deep channels. This conjecture is corroborated by the fact that the considered database largely reflects fluvial systems fed by erosion of long-lived Proterozoic orogens, such as the Trans-Hudson and Grenville. We included key sentences to highlight the point suggested by the reviewer at **lines 224-227**.